# Structural and Functional Alterations in Mitochondria-Associated Membranes (MAMs) and in Mitochondria Activate Stress Response Mechanisms in an In Vitro Model of Alzheimer’s Disease

**DOI:** 10.3390/biomedicines9080881

**Published:** 2021-07-24

**Authors:** Tânia Fernandes, Rosa Resende, Diana F. Silva, Ana P. Marques, Armanda E. Santos, Sandra M. Cardoso, M. Rosário Domingues, Paula I. Moreira, Cláudia F. Pereira

**Affiliations:** 1CNC—Center for Neuroscience and Cell Biology, CIBB—Center for Innovative Biomedicine and Biotechnology, University of Coimbra, 3004-504 Coimbra, Portugal; tfernandes@cnc.uc.pt (T.F.); rresende@cnc.uc.pt (R.R.); dianaffsilva@gmail.com (D.F.S.); apatriciabmarques@gmail.com (A.P.M.); aesantos@ci.uc.pt (A.E.S.); cardoso.sandra.m@gmail.com (S.M.C.); 2IIIUC—Institute for Interdisciplinary Research, University of Coimbra, 3030-789 Coimbra, Portugal; 3CACC—Clinical Academic Center of Coimbra, 3004-561 Coimbra, Portugal; 4Faculty of Medicine, University of Coimbra, 3000-548 Coimbra, Portugal; 5Laboratory of Biochemistry and Biology, Faculty of Pharmacy, University of Coimbra, 3000-548 Coimbra, Portugal; 6Mass Spectrometry Centre, REQUIMTE-LAQV and CESAM, Department of Chemistry, University of Aveiro, 3810-193 Aveiro, Portugal; mrd@ua.pt

**Keywords:** Alzheimer’s disease, subcellular fractions, ER-mitochondria contacts, Ca^2+^ signaling, mitochondrial dysfunction

## Abstract

Alzheimer’s disease (AD) is characterized by the accumulation of extracellular plaques composed by amyloid-β (Aβ) and intracellular neurofibrillary tangles of hyperphosphorylated tau. AD-related neurodegenerative mechanisms involve early changes of mitochondria-associated endoplasmic reticulum (ER) membranes (MAMs) and impairment of cellular events modulated by these subcellular domains. In this study, we characterized the structural and functional alterations at MAM, mitochondria, and ER/microsomes in a mouse neuroblastoma cell line (N2A) overexpressing the human amyloid precursor protein (APP) with the familial Swedish mutation (APPswe). Proteins levels were determined by Western blot, ER-mitochondria contacts were quantified by transmission electron microscopy, and Ca^2+^ homeostasis and mitochondria function were analyzed using fluorescent probes and Seahorse assays. In this in vitro AD model, we found APP accumulated in MAM and mitochondria, and altered levels of proteins implicated in ER-mitochondria tethering, Ca^2+^ signaling, mitochondrial dynamics, biogenesis and protein import, as well as in the stress response. Moreover, we observed a decreased number of close ER-mitochondria contacts, activation of the ER unfolded protein response, reduced Ca^2+^ transfer from ER to mitochondria, and impaired mitochondrial function. Together, these results demonstrate that several subcellular alterations occur in AD-like neuronal cells, which supports that the defective ER-mitochondria crosstalk is an important player in AD physiopathology.

## 1. Introduction

Alzheimer’s disease (AD) is the most common age-related neurodegenerative disorder affecting more than 47.5 million people worldwide [1]. It is characterized by the accumulation of extracellular neuritic plaques, mainly composed by amyloid β (Aβ), and intracellular neurofibrillary tangles mostly formed by hyperphosphorylated tau, as well as by progressive neuronal loss, particularly in the cerebral cortex and hippocampus, which leads to cognitive impairment [2,3]. The familial forms of AD (FAD) are caused by several mutations, including in the gene that encodes the amyloid precursor protein (APP), whose cleavage by β- and γ-secretases originates the Aβ peptide [1]. Furthermore, mutations in presenilin-1 (PS1) and presenilin-2 (PS2) that are enzymatic active components of γ-secretase complex were also associated with Aβ deposition in FAD patients [4]. The major risk factors for sporadic AD (SAD), the most prevalent form of the disease, are aging and the presence of the ℇ4 allele of apolipoprotein E (ApoE4) that, among other factors, seem to affect MAM function [5].

The disruption of MAM has been implicated in AD physiopathology [4], since it modulates several AD-related features, such as altered lipid and glucose metabolism, aberrant Ca^2+^ homeostasis, increased endoplasmic reticulum (ER) stress, and mitochondrial dysfunction, which occur years before the appearance of the pathological hallmarks of AD [6,7,8]. MAM are biochemical and physical contact sites between the ER and mitochondria with an intermembrane distance of about 10 to 80 nm [9,10,11]. This structure that exhibits the features of a lipid raft is dynamic due to the presence of a set of specialized proteins, working as an intracellular signaling platform able to determine cell fate. Indeed, the proteins present in this region regulate numerous cellular processes, such as lipid homeostasis, Ca^2+^ signaling, apoptosis, redox status, proteostasis including autophagy, and the ER stress-induced unfolded protein response (UPR), as well as mitochondrial dynamics and bioenergetics [12,13,14,15,16]. Mitochondria are conserved organelles that play an important role in neuronal cell fate because they regulate both the energy metabolism and cell death pathways. Due to their essential role in energy production, among other things, mitochondrial dysfunction is also considered an early disease feature in vulnerable neurons of the brains of AD patients [17,18,19]. 

The main goal of our study was to investigate the structural alterations at MAM using an in vitro model of AD, namely the mouse neuroblastoma cell line (N2A) overexpressing the APP familial Swedish mutation (APPswe). The impact of MAM alterations on several cellular stress responses as well as in mitochondria functioning was also evaluated, including Ca^2+^ transfer from the ER, energetic metabolism and ATP production, dynamics, and biogenesis concomitantly with the analysis of ER stress-induced UPR. 

## 2. Materials and Methods

In order, to study the structural alterations at MAM and the impact of these on cellular stress responses and mitochondria function in AD, we used an in vitro cell model. We isolated MAM, microsomes, and mitochondria fractions from N2A-WT and N2A-APPswe cell lines and protein levels of ER-mitochondria tethers, Ca^2+^ signaling and stress response mediators were determined by Western blot (WB). ER-mitochondria contacts were quantified by transmission electron microscopy, and Ca^2+^ homeostasis and mitochondria function were analyzed using fluorescent probes and Seahorse assays, respectively.

### 2.1. Cell Culture

The wild-type mouse neuroblastoma cell line N2A (WT) and the N2A-APPswe cell line, which stably overexpress human Swedish mutant APP KM670/671NL (APPswe) [20], were maintained in Dulbecco’s modified Eagle’s medium (DMEM, Sigma-Aldrich, St. Louis, MO, USA, #D5648) supplemented with 10% (*v/v*) fetal bovine serum (FBS) (Gibco, Waltham, MA, USA, #26400-044), 3.7 g/L sodium bicarbonate (Merck, Kenilworth, NJ, USA, S8761), 1% (*v/v*) non-essential amino acids (Merck, #M7145), and 1 mM sodium pyruvate (Merck, #S8636), as previously described [21,22]. Cell culture medium was also supplemented with 1% (*v/v*) penicillin/streptomycin (Gibco, #15140-122) or 0.4 mg/mL geneticin (Gibco, #10131027) for WT or APPswe cells, respectively. Both cell lines were cultured at 37 °C in a humidified 5% CO_2_–95% air atmosphere. 

### 2.2. Subcellular Fractionation

Purification of microsomes, MAM, and crude mitochondria was performed using modifications of the protocols previously described by Wieckowski et al. [23] and Williamson et al. [24]. Cells were harvested and centrifuged at 200× *g* for 5 min at 4 °C. Homogenization of pellets was then performed gently with a glass/teflon homogenizer in isolation buffer, pH 7.4, composed of 250 mM sucrose and 10 mM HEPES for microsomes and MAM isolation, or 225 mM mannitol, 75 mM sucrose, 30 mM Tris-HCl, and 0.1 mM EGTA for crude mitochondria isolation, supplemented with a 1% cocktail of proteases inhibitors (Sigma-Aldrich, #P2714). The suspension was centrifuged eight times at 600× *g* for 5 min at 4 °C to remove cell debris and nuclei and obtain the total fraction (TF). For microsomes and MAM isolation, the TF was centrifuged at 10,300× *g* for 10 min at 4 °C, resulting in a supernatant (cytosolic and ER/microsomes fraction) and pellet (MAM fraction). The supernatant was centrifuged at 100,000× *g* for 60 min at 4 °C in a Beckman ultracentrifuge (Indianapolis, IN, USA, model L-100 XP, 90 Ti rotor) to pellet the ER/microsomal fraction that was then resuspended in PBS. For MAM isolation, the pellet resulting from TF centrifugation was resuspended in 1 mL SHM solution (250 mM sucrose and 10 mM HEPES) and centrifuged at 10,300× *g* for 10 min at 4 °C. The resulting pellet was resuspended in 600 µL mannitol buffer A (250 mM mannitol, 0.5 mM EGTA, and 5 mM HEPES, pH 7.4), loaded on top of a 30% (*v/v*) Percoll gradient, and centrifuged for 65 min at 95,000× *g* in a Beckman ultracentrifuge (Indianapolis, IN, USA, model L-100 XP, SW41 rotor). The upper band (containing the MAM fraction) was collected, diluted in PBS containing 1 mM PMSF, and centrifuged at 6300× *g* for 10 min at 4 °C. The supernatant was centrifuged at 100,000× *g* for 30 min at 4 °C in a Beckman centrifuge (Indianapolis, IN, USA, model Avanti J-26 XPI, JA 25.15 rotor), and the pellet (MAM purified fraction) was resuspended in PBS containing 1 mM PMSF. To obtain the crude mitochondria fraction, the TF was centrifuged at 7000× *g* for 10 min at 4 °C in a Beckman centrifuge (model Avanti J-26 XPI, JA 25.15 rotor), the pellet was resuspended in starting buffer (225 mM mannitol, 75 mM sucrose, and 30 mM Tris-HCl, pH 7.4), and centrifuged at 7000× *g* for 10 min at 4 °C. This procedure was repeated five times. The mitochondrial pellet was resuspended in starting buffer, centrifuged at 10,000× *g* for 10 min at 4 °C in a Beckman centrifuge (model Avanti J-26 XPI, JA 25.15 rotor), and the resulting pellet (crude mitochondrial fraction) was resuspended in MRB buffer (250 mM mannitol, 5 mM HEPES, and 0.5 mM EGTA, pH 7.4). Subcellular fractions purity was analyzed by WB (Appendix A).

### 2.3. Protein Analysis by Western Blot

Protein concentration was determined with the Pierce BCA Protein Assay Kit (Thermo Fisher, Waltham, MA, USA, #23227). Loading buffer was added to samples that were then boiled and denatured. Sample proteins (30–50 µg) were separated using 7.5–15% SDS-PAGE, transferred to a PVDF membrane (Millipore, Burlington, MA, USA), and blocked in 5% (*w/v*) BSA Tris-buffered saline containing 0.1% (*v/v*) Tween-20 (TBS-T). Membranes were incubated with the primary antibody overnight at 4 °C and afterward they were washed with TBS-T and incubated with the secondary antibody at room temperature for 1 h. Membranes were developed using ECF substrate (GE Healthcare, Chicago, IL, USA, #RPN5785) in ChemiDoc Imaging System (Bio-Rad, Hercules, CA, USA), and quantifications were performed with the Bio-Rad Image Lab Software 6.1. Primary and secondary antibodies utilized for WB are summarized in Table 1 and Table 2, respectively.

### 2.4. Mitochondria Morphology Analysis by Transmission Electron Microscopy (TEM) 

WT and APPswe cells were collected and centrifuged at 1008× *g* for 5 min to form a pellet. Cells were fixed with 2.5% (*w/v*) glutaraldehyde in 0.1 M sodium cacodylate buffer (pH 7.2) for 2 h. Afterwards, cells were washed in the same buffer and the post-fixation was performed using 1% (*w/v*) osmium tetroxide for 1 h. Then, the cell pellets were rinsed two times with buffer and distilled water and, for contrast enhancement, cells were incubated in 1% (*w/v*) aqueous uranyl acetate during 1 h. After washing with water, cell pellets were embedded in 2% (*w/v*) molten agar and dehydrated in ethanol (30–100%). Then, cell pellets were impregnated and included in Epoxy resin (Fluka Analytical, Charlotte, NC, USA). After polymerization, ultrathin sections were obtained, and observations were carried out on a FEI-Tecnai G2 Spirit Bio Twin at 100 kV. We analyzed approximately 10 different cells per cell line (*n* = 3) and MAM was considered when the distance between ER and mitochondria, which was measured using the ImageJ program, was ≤25 nm. At this distance, Ca^2+^ transfer between both organelles occurs through the IP_3_R-GRP75-VDAC axis [10]. The number of ER-mitochondria contacts ≤ 25 nm were obtained dividing the number of MAM per number of total mitochondria. 

### 2.5. Mitochondria Morphology Analysis by Confocal Microscopy Using MitoTracker Green 

For imaging experiments, cells were cultured on 18 mm glass bottom culture 12-well plate (150,000 cells per well) coated with poly-lysine D and incubated at 37 °C in a humidified 5% CO_2_–95% air atmosphere for 24 h. Culture medium was discarded, cells washed in Krebs solution (140 mM NaCl, 5 mM KCl, 1.5 mM CaCl_2_, 1 mM MgCl_2_, 1 mM NaH_2_PO_4_, 9.6 mM glucose, 20 mM HEPES; pH 7.4), and loaded with 100 nM MitoTracker Green probe (Invitrogen, Waltham, MA, USA, #M7514) in Krebs solution for 30 min at 37 °C in a humidified 5% CO_2_. Then, cells were washed in Krebs solution and incubated with 15 µg/mL Hoechst 33342 (Molecular Probes) in Krebs solution for 5 min at same conditions. Cells were washed and live images of the cells (608 WT cells and 498 APPswe cells) were captured with the Zeiss LSM 710 Confocal Microscope (Carl Zeiss, Jena, Germany) with 63× oil objective and the fluorescence intensity analyzed using ImageJ software. The value of corrected total cell fluorescence (CTCF) was calculated using the following formula: CTCF = integrated density of selected cell − (area of selected cell × mean fluorescence of background readings).

### 2.6. Fluorometric Analysis of Calcium and Mitochondrial Membrane Potential

Intracellular cytosolic Ca^2+^ levels were measured with Fura-2/AM (Invitrogen, #F1221) and mitochondrial Ca^2+^ content was measured with Rhod-2/AM (Invitrogen, #R1244). Cells were cultured in 48-well plates (75,000 cells per well) in triplicate and incubated at 37 °C in a humidified 5% CO_2_–95% air atmosphere. The next day, cells were washed with solution A (140 mM NaCl, 2.5 mM KCl, 1 mM MgCl_2_, 20 mM HEPES, 10 mM glucose, and 1.8 mM CaCl_2_, pH 7.4) and incubated for 30 min with 2 µM Fura-2/AM in solution A supplemented with 0.1% (*w/v*) BSA or for 45 min with 10 µM Rhod-2/AM in solution A at 37 °C. Cells were rinsed with solution B (140 mM NaCl, 2.5 mM KCl, 1 mM MgCl_2_, 20 mM HEPES, and 10 mM glucose, pH 7.4) and Ca^2+^ levels were measured upon stimulation with 5 µM thapsigargin (Sigma-Aldrich, #T9033) or 100 µM histamine (Sigma-Aldrich, #H7125). The fluorescence signal was measured at 37 °C during 10 min (10 in 10 sec) at λEx = 340/380 nm and λEm = 510 nm for Fura-2/AM and λEx = 552 nm and λEm = 581 nm for Rhod-2/AM using a plate reader (Fluorimeter SpectraMax Gemini EM, Molecular Devices, San Jose, CA, USA). 

Mitochondrial membrane potential (Δψm) was evaluated using the fluorescent probe TMRE (tetramethylrhodamine ethyl ester perchlorate). Cells were seeded in 96-well plates (25,000 cells per well) in triplicate and incubated at 37 °C in a humidified 5% CO_2_–95% air atmosphere for 24 h. Culture medium was discarded and replaced by fresh medium containing 4 µM TMRE (Sigma-Aldrich, #87917) and cells were incubated at 37 °C for 30 min. Subsequently, the supernatant was removed, cells were washed using PBS and the fluorescence signal (λEx = 544 nm; λEm = 590 nm) was measured at 37 °C using a plate reader (Fluorimeter SpectraMax Gemini EM, Molecular Devices, San Jose, CA, USA). 

The protein content of each well was determined by aspirating the assay buffer, followed by addition of 10 and 20 µL RIPA buffer (250 mM NaCl, 50 mM Tris, 1% Nonidet P-40, 0.5% DOC, and 0.1% SDS) for TMRE and Ca^2+^ assays, respectively. The multi-well plates were placed on a plate shaker at low speed for 10 min and then on ice during 20 min. Finally, 100 and 200 µL BCA assay mix (Invitrogen, #23227) were added for the TMRE and Ca^2+^ assays, respectively. After a 30 min incubation period, absorbance was measured at 562 nm with a Spectrofotometer Spectramax plus 384 (Molecular Devices, San Jose, CA, USA). 

### 2.7. Activity of Mitochondrial Electron Transport Chain and Glycolysis Measured by the Seahorse XFe24 Analyzer

Cells were seeded (50,000 cells per well) in cell culture microplates (5 wells per cell line) provided by the manufacturer and incubated at 37 °C in a humidified 5% CO_2_–95% air atmosphere. The next day, cells were washed in unbuffered DMEM (Sigma-Aldrich, #5030) supplemented with 25 mM glucose, 1 mM sodium pyruvate, 0.58 g/L L-glutamine, and 1% (*v/v*) non-essential amino acids for oxygen consumption rate (OCR) assay or in unbuffered glucose-free DMEM for extracellular acidification rate (ECAR) assay. Cells were incubated for 1 h at 37 °C in pre-warmed medium with pH adjusted to 7.4 and plates were loaded into a Seahorse XF24 (Argilent, Santa Clara, CA, USA). OCR was determined under the following conditions: Basal measurements; after 1 µM oligomycin (Alfa Aesar, Karlsruhe, Germany, #J60211) addition; upon injection of 1 µM FCCP (Sigma Chemical, #C2920) and, finally, after a mixture of 2 μM rotenone (Sigma Chemical, #R8875) plus 2 μM antimycin A (Sigma Chemical, #A8674) was injected. Rotenone (a mitochondrial complex I inhibitor) and antimycin A (a mitochondrial complex III inhibitor) inhibit mitochondrial respiration allowing to determine the non-mitochondrial OCR. To calculate the basal mitochondrial OCR, the non-mitochondrial OCR is subtracted from the OCR obtained before the addition of oligomycin (inhibits the mitochondrial ATP synthase). To evaluate the maximal respiratory capacity, we subtracted the non-mitochondrial OCR from the OCR following carbonyl cyanide-4-(trifluoromethoxy) phenyl hydrazone (FCCP) addition (disrupts ATP synthesis by dissipating the proton gradient). To obtain the spare respiratory capacity, we subtracted the basal mitochondrial OCR from the maximal respiratory capacity. The mitochondrial ATP production was assessed after oligomycin addition, by subtracting the non-mitochondrial OCR from the OCR value after oligomycin.

In the ECAR assay, after baseline measurements, the following injections were performed to each well under similar conditions: First, 25 mM glucose, then 1 µM oligomycin, and, finally, an injection of 100 mM 2-deoxyglucose (2DG) provided a non-glycolysis extracellular acidification rate. Cells were first placed in a glucose-free medium and the first injection consisted in 25 mM glucose. The glycolysis rate was obtained by subtracting non-glycolysis ECAR from the ECAR post-glucose. Then, oligomycin was added to the cells to suppress mitochondrial ATP production. The glycolysis capacity was determined by the difference between the post-oligomycin ECAR and post-2-deoxyglucose baseline ECAR. After the assays, buffer was aspirated, 30 µL RIPA buffer was added per well, and protein content was measured using BCA assay mix to determine the protein content per well. In comparison with the basal rates, the % of change was calculated dividing the measured value by the average value of baseline readings and normalized to the total protein content of each well. The OCRs are normalized to the WT cells mean.

### 2.8. Statistical Analysis

Statistical analysis was performed using the GraphPad Prisma 8.0.2 software (San Diego, CA, USA) and presented as mean ± standard error of the mean (SEM). All experiments were independent assays. Data were tested for Gaussian distribution using the Shapiro-Wilk test. Differences between the two groups were analyzed using unpaired *t*-test and two-way ANOVA with Sidak post hoc correction for grouped analysis. In case of non-normality distribution, data were analyzed by the Mann–Whitney U-test to compare two groups and Kruskal–Wallis Dunn’s correction for multiple comparisons. Statistical significance was considered at *p* < 0.05.

## 3. Results

### 3.1. Alterations of ER-Mitochondria Contacts in APPswe Cells

The direct contact between organelles allows the exchange of signals and metabolites that have a role in cell physiology and stress response [25]. The ER-mitochondria contacts are dynamic structures, with the distance between the two membranes ranging from 10 to 80 nm [9,10]. The formation of these contacts requires the presence of tethering proteins inserted in the outer mitochondrial membrane (OMM) that interact with ER membrane-resident proteins. The Ca^2+^ channel, associated with the inositol 1,4,5-trisphosphate receptor (IP_3_R) present at ER, interacts with the mitochondrial voltage-dependent anion channel 1 (VDAC1) through the glucose-regulated protein 75 (GRP75) and mitofusin 2 (MFN2), present in both ER and mitochondria homo- and hetero-oligomerizes with the mitochondrial mitofusin 1 (MFN1), to tether both organelles [26,27,28]. In our study, the ER-mitochondria tethering proteins MFN1 and MFN2 were analyzed by WB in microsomes (heterogenous set of vesicles formed from the ER), mitochondria, and MAM subcellular fractions isolated from WT and APPswe cells. We observed an increase in MFN1 protein levels in MAM fraction (*p* < 0.0794) in APPswe cells when compared with WT cells (Figure 1a,b) and, as expected, this protein was not detected in the microsomes. Furthermore, MFN2 content was significantly decreased in whole-cell extracts and isolated subcellular fractions (microsomes, mitochondria, and MAM) in the APPswe compared with WT cells (total fraction: *p* < 0.0001, microsomes: *p* < 0.001, mitochondria: *p* < 0.05, MAM: *p* < 0.05) (Figure 1a,c). Since this data suggests alterations in ER-mitochondria tethering, we analyzed the contact between both organelles using transmission electron microscopy (TEM) that allows the direct visualization of MAM. Tethers identified in TEM span from 5 to 80 nm. We observed a mild decrease of ER-mitochondria contacts ≤ 25 nm (*p* = 0.0656) (Figure 1d,e) and a significant decrease of mitochondria per cell (*p* < 0.001) (Figure 1f) in APPswe when compared with WT cells. Using the MitoTracker Green probe, we also evaluated the mitochondrial mass/abundance by confocal microscopy. A significant decrease of approximately 25% (*p* < 0.0001) in mitochondria fluorescence intensity was observed in APPswe cells when compared with WT cells (Figure 1g,h). Additionally, we observed a perinuclear localization of mitochondria in APPswe cells, similarly to what has been described for other AD models [29,30]. Altogether these observations indicate a decrease in ER-mitochondria contacts in APPswe cells and, consequently, a possible reduction in the communication between both organelles that affects cell homeostasis.

### 3.2. Impairment of Ca^2+^ Transfer from ER to Mitochondria in APPswe Cells

Mitochondrial metabolism is versatile and capable to adjust to specific physiological or pathological conditions regulating fundamental cell pathways that range from proliferation to apoptosis [15]. The functional interaction between mitochondria and ER influences mitochondrial Ca^2+^ signaling, which in turn regulates mitochondrial function. Dysregulation of mitochondrial Ca^2+^ and metabolism have been associated with AD development [19,31,32]. Cytosolic Ca^2+^ levels were measured with the Fura-2/AM fluorescent probe following treatment with thapsigargin (TG), which blocks the ability of the cells to pump Ca^2+^ into the ER by inhibiting the sarco/endoplasmic reticulum Ca^2+^ ATPase (SERCA) and depletes ER Ca^2+^. We observed a significant increase in Ca^2+^ flux from ER to cytosol (*p* < 0.01) in APPswe cells (Figure 2a,b), suggesting a higher ER Ca^2+^ content in these cells in comparison with WT cells. The width of the ER-mitochondria cleft influences Ca^2+^ transfer through the IP_3_R-GRP75-VDAC complex, which assembles when the distance between both organelles is in the range of 10–25 nm [10]. Since our results show a decrease in ER-mitochondria contacts ≤ 25 nm in the APPswe cells, we also examined mitochondrial Ca^2+^ levels using the Rhod-2/AM probe. We observed a statistically significant decrease (approximately 50%) in Ca^2+^ flux from ER to mitochondria (*p* = 0.05) in APPswe cells following treatment with histamine that triggers Ca^2+^ release through IP_3_R (Figure 2c,d). Then, we analyzed the levels of VDAC1 and mitochondrial Ca^2+^ uniporter (MCU) proteins by WB. VDAC1 is a protein present in the OMM involved in ER-mitochondria tethering by forming a ternary bridging complex with the OMM chaperone GRP75 and IP_3_R in the ER, as well as in Ca^2+^ signaling by regulating Ca^2+^ transfer from ER to mitochondria via IP_3_R [28,33]. To enter in mitochondrial matrix, Ca^2+^ must cross the MCU located in the inner mitochondrial membrane (IMM) [34]. A significant increase of VDAC1 content in mitochondrial (*p* < 0.05) and MAM (*p* < 0.05) fractions was observed in APPswe cells when compared with WT cells (Figure 2e,g). Furthermore, we detected a significant decrease in MCU levels in APPswe whole-cell extracts (*p* < 0.01) (Figure 2f,i), which was not observed in the mitochondria fraction. Besides that, we also observed a significant increase of sigma-1 receptor (Sig-1R) levels in MAM fraction (*p* < 0.05) (Figure 2f,j), and a significant decrease of glucose-regulated protein 78 (BiP/GRP78) expression in whole-cell extracts (*p* < 0.05) and microsomes (*p* < 0.01) isolated from APPswe cells (Figure 2e,h). Sig-1R is a chaperone that resides mainly at MAM, which is involved in the structural integrity of the MAM and plays a role in the regulation of Ca^2+^ signaling regulation between ER and mitochondria by coupling to the IP_3_R. Under ER stress, Sig-1R dissociates from the co-chaperone BiP/GRP78 and acts as a free chaperone to stabilize IP_3_Rs, leading to an increased Ca^2+^ transfer from ER to mitochondria that promote ATP production [34,35]. Despite the alterations in ER Ca^2+^ levels and the increased protein levels of VDAC1 and Sig-1R observed in APPswe cells, a decrease in Ca^2+^ flux from ER to mitochondria was found in these cells that can arise from the decrease in ER-mitochondria contacts, as well as due to changes in mitochondria membrane potential. 

### 3.3. Oxygen and Glucose Fluxes in APPswe Cells

Mitochondria have an essential role in the bioenergetic process and are involved in amino acid, lipid, and steroid metabolism, Ca^2+^ homeostasis, ATP production via oxidative phosphorylation (OXPHOS), and apoptosis. Thus, these organelles regulate cellular fate and mitochondrial dysfunction is one of the most prominent features in AD that occurs in the early stage of this disease [17,18,36,37,38]. Because neuronal survival is dependent on mitochondrial function that, in turn, is affected by Ca^2+^ flux from the ER, which we showed to be decreased in APPswe cells (Figure 2d), we assessed mitochondrial function parameters, namely OCR (Figure 3a), ATP production levels, and mitochondrial membrane potential. We observed that the basal mitochondrial respiration, determined by monitoring ORC in the absence of any inhibitors, was significantly decreased in APPswe cells when compared with WT cells. (*p* < 0.01) (Figure 3b). Additionally, we found that maximal respiratory capacity, which indicates the maximal oxygen consumption rate attained by adding the uncoupler FCCP, was significantly decreased in APPswe cells (*p* < 0.001) (Figure 3c), showing that mitochondria from mutant cells are already working at an increased rate. Accordingly, the spare respiratory capacity, described as the amount of extra ATP that can be produced by OXPHOS in case of a sudden increase in energy demand, was also significantly decreased in APPswe cells when compared with WT cells (*p* < 0.001) (Figure 3d). In the presence of oligomycin, an inhibitor of ATP synthase, mitochondrial oxygen consumption is due to ATP synthase-independent leakage of protons from the intermembrane space to the mitochondrial matrix. Under these conditions, we observed a 30% reduction in ATP production (*p* < 0.05) in APPswe cells (Figure 3e).

Since alterations in mitochondrial respiration are associated with changes in mitochondrial membrane potential, we used the TMRE fluorescence probe to assess this mitochondrial parameter. In line with the OCR data, APPswe cells showed mitochondrial membrane depolarization when compared with WT cells (*p* < 0.01) (Figure 3f). Altogether, these results show that mitochondria are significantly impaired in APPswe cells, which suggest a suppression in mitochondrial activity. However, this result might be, at least in part, caused not only by biochemical alterations but by the reduction in mitochondrial mass/abundance that we observed (Figure 1h).

Our studies demonstrated an impairment of mitochondria function that resulted in a decline in the production of ATP via OXPHOS; therefore, we measured glycolysis to study if this pathway could compensate the decrease in ATP synthesis by OXPHOS in APPswe cells. Glycolysis fluxes can be estimated by measuring the pH of cell culture medium since lactic acid, the end-product of anaerobic glycolysis, contributes to its acidification. At the end of the experiments, 2-deoxyglucose (2DG) was added to determine the non-glycolysis acidification rate (Figure 3g). We detected, in APPswe cells, a 40% reduction in glycolysis rate determined by the quantification of the extracellular acidification rate (ECAR) of the surrounding media (*p* < 0.01) (Figure 3h) and a 30% reduction in glycolytic capacity, evaluated by the difference between the post-oligomycin ECAR and the post-2-deoxyglucose baseline ECAR (*p* < 0.01) (Figure 3i) when compared with WT cells. These results suggest that, in this in vitro AD model, cells present an energy deficit resulting from defects in both mitochondrial respiration and glycolytic pathway.

### 3.4. Impaired Mitochondrial Dynamics and Biogenesis in APPswe Cells

Mitochondria are dynamic organelles undergoing cycles of fusion (union of two mitochondria) and fission (division of one mitochondrion into two daughter mitochondria) to maintain their shape, distribution, and size. A balanced mitochondrial dynamics is required to ensure a proper mitochondrial function and response to cellular stress [39]. Impaired fusion-fission balance has a key role in mitochondrial dysfunction and can cause neuronal dysregulation, being associated to several neurological disorders [18]. We analyzed by WB the levels of proteins involved in fusion, MFN1 and MFN2, and fission, dynamin-1-like protein (DRP1), and mitochondrial fission 1 (FIS-1). The quantification analysis in total extracts revealed a statistically significant decrease in MFN2 levels (*p* < 0.05) (Figure 4a,d) in the absence of relevant alterations in MFN1, *p*-DRP1/DRP1 and FIS-1 (Figure 4a–c,e) in APPswe cells when compared with WT cells, suggesting an impaired mitochondrial dynamics in APPswe cells.

Mitochondrial biogenesis is a process by which mitochondria increase their number/mass [40] contributing to mitochondrial and cellular homeostasis [18,36]. We analyzed the levels of factors that regulate mitochondrial biogenesis, namely nuclear respiratory factor 1 (NRF1), which controls nuclear genes that encode the mitochondrial proteins, NADH-ubiquinone oxidoreductase chain 1 (ND1), which is involved in the transfer of electrons from NADH to the respiratory chain, and the mitochondrial transcription factor A (mtTFA), which drives transcription and replication of mitochondrial DNA. The WB results revealed that the levels of these proteins in total cellular extracts are slightly increased in APPswe cells when compared with WT cells (Figure 4f–h), but the differences are not statistically significant. These observations suggest a compensatory mechanism in APPswe cells to face the decline of mitochondrial function and mass (Figure 1g,h). 

### 3.5. Cellular Stress Responses Are Affected in APPswe Cells

ER is the main compartment involved in protein synthesis and folding as well as Ca^2+^ signaling, and upon stress activates the UPR to restore proteostasis. The UPR activation can be triggered by the accumulation of pathogenic misfolded proteins, disruption of intracellular Ca^2+^ homeostasis, defects in autophagy, oxidative stress, proteasome inhibition, and metabolic or mitochondrial dysfunctions [41,42]. Protein kinase R-like ER kinase (PERK) and inositol-requiring enzyme 1α (IRE1α) are ER stress sensors that are inactive when linked to the chaperone BiP/GRP78 [41]. Upon ER stress, the interaction between BiP/GRP78 and sensor proteins is inhibited and UPR signaling is initiated [42]. PERK is a transmembrane kinase located at ER membrane and its activation inhibits protein synthesis. IRE1α is a transmembrane kinase and endoribonuclease that regulates the expression of genes involved in protein folding, ER-associated degradation (ERAD), protein quality control, and organelle biogenesis. The ER oxidase 1 (ERO1α) plays an important role in ER redox potential and its activity is central to maintain cell redox homeostasis, since it oxidizes the active-site cysteines of protein disulfide isomerase (PDI), an essential enzyme that is responsible for the addition of disulfide bonds into newly synthesized proteins. This protein is also involved in Ca^2+^ homeostasis and MAM signaling and its downregulation inhibits mitochondrial Ca^2+^ uptake [43,44]. In the in vitro model of AD used in this study, human APP with the Swedish mutation is overexpressed and Aβ is generated [20,45], which can induce ER stress and consequently activate the ER stress response [46,47]. Using WB analyses, we observed a significant accumulation of APP in mitochondria (*p* < 0.0001) and MAM fractions (*p* < 0.05) (Figure 5a,b). Due to this evidence, we evaluated UPR activation measuring by WB the protein levels of PERK, IRE1α, and ERO1α. A slight increase in PERK levels were detected in total extracts, microsomes, and MAM subcellular fractions (Figure 5c,d) and a significant IRE1α upregulation was found in MAM fraction (*p* < 0.05) in APPswe cells (Figure 5c,e). Moreover, the content of ERO1α was significantly decreased in all subcellular fractions (total fraction: *p* < 0.0001, microsomes: *p* < 0.05, mitochondria: *p* < 0.0001, MAM: *p* < 0.0001) obtained from APPswe cells when compared with WT cells (Figure 5f, g), which could lead to the accumulation of misfolded proteins in the ER lumen. Finally, APPswe cells presented a significant increase of proliferating cell nuclear antigen (PCNA) protein levels in all subcellular fractions (total fraction: *p* < 0.01, microsomes: *p* < 0.01, mitochondria: *p* < 0.05, MAM: *p* < 0.001) when compared with WT cells (Figure 5f,h). PCNA is an essential cofactor for DNA replication and repair, thus being a cell lifespan regulator [48]. Together, these results suggest that protein accumulation in APPswe cells, as well as Ca^2+^ deregulation and mitochondrial dysfunction, induce ER stress and activate the UPR, which subsequently upregulates PCNA to maintain cells viability.

The pore translocase of the outer mitochondrial membrane 40 (TOM40) is essential for mitochondrial functioning, since it is required for protein transport into mitochondria, including for the internalization of APP and Aβ [49]. Accumulation of misfolded and aggregated proteins in the mitochondria matrix activates the mitochondrial UPR, where the chaperone heat shock protein 60 (Hsp60) plays an essential role to preserve protein homeostasis [50]. Our WB analyses revealed a significant increase of TOM40 in mitochondria (*p* < 0.0001) and MAM (*p* < 0.001) fractions (Figure 5f,i) and of Hsp60 in MAM fraction (*p* < 0.01) (Figure 5j,k) in APPswe cells when compared with WT cells. These results suggest that there is an increased transport of APP to mitochondria through the TOM complex, and an accumulation of unfolded/misfolded proteins in this organelle increase the level of the mitochondrial UPR Hsp60 in MAM fraction.

## 4. Discussion

It has been described that MAM-related functions are perturbed in several neurodegenerative disorders, such as AD, Parkinson’s disease and amyotrophic lateral sclerosis [4]. In AD, an aberrant Ca^2+^ regulation, increased ER stress, alteration in glucose metabolism, and mitochondrial dysfunction have been reported. These alterations may, in part, be attributed to the disruption of the MAM-related functions [6,51,52]. 

In the present study we characterized the structural and functional alterations in MAM and mitochondria and their effect on stress response mechanisms, in an in vitro model of AD, to support the role of MAM in AD and their potential as therapeutic targets in this disease. Overall, our findings demonstrate: (1) Decreased ER-mitochondria contacts and alteration of MAM’s composition; (2) changes in ER-mitochondria Ca^2+^ transfer; (3) alterations in mitochondrial dynamics and function. 

First, we analyzed the structural and functional alterations at MAM in APPswe and WT cells. APPswe cells recapitulate amyloid pathology through increased Aβ generation [20]. ER-mitochondria contacts require the interaction of tethering proteins present in the ER membrane and OMM, such as MFN1 and MFN2 [27]. In APPswe cells, we observed a slight increase of MFN1 protein levels in mitochondria and MAM fractions (Figure 1a,b) and a decrease of MFN2 levels in all fractions, namely microsomes, mitochondria, and MAM fractions (Figure 1a,c). Previous studies showed that mitochondrial anomalies, oxidative stress, inflammation, and cytoskeletal rearrangements contribute to neurodegeneration in the hippocampus and cortex of a MFN2 KO mouse model [53,54]. Park et al. also found a decrease in MFN2 expression in neuro-2a cells expressing the Swedish mutation of APP [55], and similar observations were also reported in human AD brains [29,56]. A decrease in the number of ER-mitochondria contacts ≤ 25 nm was observed in APPswe cells (Figure 1e), which can be associated with the decreased content of MFN2 (Figure 1a,c). Accordingly, it has been previously found that MFN2 ablation or silencing in mouse embryonic fibroblasts and HeLa cells, respectively, reduce the ER-mitochondria juxtaposition affecting mitochondrial uptake of Ca^2+^ released by the ER [57,58]. The reduction of ER-mitochondria contacts was also reported in senescent cells [59]. We also found a decrease in the number of mitochondria per cell (Figure 1f) and in mitochondrial mass (Figure 1h) in APPswe cells. Jiang et al. observed that neurons of MFN2 knockout mice present a decreased number of mitochondria in the axons and dendrites, changes in mitochondrial morphology, and fragmentation caused by an altered mitochondrial fusion [53]. Possibly, the decrease of ER-mitochondria contacts is associated with the reduction of Ca^2+^ flux, since we observed a decrease in mitochondrial Ca^2+^ levels after histamine stimulation (Figure 2d), despite the accumulation of Ca^2+^ in the ER (Figure 2b). Concerning mitochondrial Ca^2+^ content, discrepancies between studies are found in literature, which may be due to the use of different models of AD (in vitro and in vivo), as well as different in vitro experimental conditions (different models and incubation period, methodologies used to determine these parameters, among others). Other authors demonstrate that mitochondrial Ca^2+^ overload and, consequently, cell death, is induced by Aβ oligomers [60,61]. Calvo-Rodrigues et al. also reported that Aβ aggregates are associated with an increase in the cytosolic as well as mitochondrial Ca^2+^ levels via MCU in the APP/PS1 transgenic (Tg) mouse model. The mitochondrial Ca^2+^ overload can induce to the opening of the mitochondrial permeability transition pore (mPTP) and caspases activation, consequently leading to cell death. The same authors reported that blocking MCU with Ru360 prevents the increase of mitochondrial Ca^2+^ levels [31], which is in accordance with a previous study in MCU knockdown HeLa cells that show an inhibition of mitochondrial Ca^2+^ uptake, even in the absence of mitochondrial membrane depolarization [62], suggesting that MCU plays a key role in mitochondrial Ca^2+^ influx. Moreover, human post-mortem AD brains show a downregulation of genes related to mitochondrial influx Ca^2+^ transporter, *MCU*, and its regulatory subunits, and an upregulation of genes involved in mitochondrial Ca^2+^ efflux pathways, *SLC8B1* (encoding NCLX), suggesting an adaptive mechanism to prevent mitochondrial Ca^2+^ overload [31]. A reduction in mitochondrial Ca^2+^ uptake in Mfn2^−/−^ MEFs in response to IP_3_-mediated cytosolic Ca^2+^ transients was also reported [57,58]. Furthermore, MCU expression was reduced by about 50% in Mfn2^−/−^ MEFs as compared with control cells [63]. However, other authors suggest that MFN2 knockdown increases ER-mitochondria contacts length and the Ca^2+^ transfer between both organelles [63,64]. In MFN2 knockdown cells, Leal et al. also observed an increase in MFN1 protein expression [64], which can compensate the MFN2 downregulation in terms of energy production and cell viability. The efficiency of ER-mitochondria Ca^2+^ transfer depends not only on the distance between both organelles, but also on the expression level of mitochondrial Ca^2+^ uptake machinery [15] and the mitochondrial membrane potential [65]. We found an increase of VDAC1 (in mitochondria and MAM fraction) (Figure 2e,g) and Sig-1R (in MAM fraction) (Figure 2f,j) levels, together with a decrease of MCU (in total extracts) (Figure 2f,i) and BiP/GRP78 levels (in total extracts and microsomes fraction) (Figure 2e,h) in APPswe cells. VDAC1 regulates ER-mitochondria Ca^2+^ signaling, and Aβ is capable to upregulate VDAC1 in human neuroblastoma cell line [66]. Its overexpression was also observed in human post-mortem AD brains and in APP Tg mice in an age-dependent manner [66]. Sig-1R resides at the MAM, forms a Ca^2+^-sensitive chaperone complex with BiP/GRP78, and associates with IP_3_R. Upon IP_3_R activation, which decreases Ca^2+^ concentration at MAM, Sig-1R dissociates from BiP/GRP78 in ER periphery, promoting Ca^2+^ signaling into mitochondria via IP_3_R [34,67]. Sig-1R up-regulation was found in APPSwe/Lon mouse brains before amyloid deposition, while decreased Sig-1R expression was observed in human post-mortem brain cortical tissue [68]. Sig-1R upregulation can also repress cell death signals in HEK293 and WT cells under ER stress [69]. Despite the decrease of BiP/GRP78 expression in APPswe cells, we also observed an accumulation of this protein in mitochondria, suggesting that BiP/GRP78 is relocated to the mitochondria (Figure 2e). A previous study shows the translocation of BiP/GRP78 to the mitochondria under ER stress that can be involved in UPR signaling between ER and mitochondria, which regulates cellular Ca^2+^ homeostasis [70]. Simultaneous, decreased BiP/GRP78 and increased in protein ubiquitination levels was observed in the brains of old rodents suggesting an impairment of ER stress response [71,72]. A decreased expression of BiP/GRP78 mRNA was also observed in primary cultures of neurons from mice with a knock-in PS1 mutant allele and in the brains of FAD and SAD patients [73]. However, other authors suggest that BiP/GRP78 expression is increased in “healthy” neurons from the human AD temporal cortex and hippocampus, which do not co-localize with neurofibrillary tangles [74]. An increase in BiP/GRP78 expression levels associated with the accumulation of toxic Aβ peptide was also observed in 2 month-old 3xTg-AD mice [75] and hippocampal cultures [76]. Under thapsigargin-induced ER stress BiP/GRP78 is bound to immature and unfolded APP preventing its translocation to distal compartments, which results in a reduction of Aβ generation because β-/γ-secretases activity is believed to be located in a distal compartment of the ER [77,78]. This evidence suggests that BiP/GRP78 can play a neuroprotective role in the initial phase of AD, but with the progression of the disease its decreased levels can attenuate the UPR signaling pathway resulting in an accumulation of unfolded proteins, namely APP. A reduction in the uptake of Ca^2+^ to mitochondria affects their metabolism contributing to mitochondria dysfunction [15]. Accordingly, we observed a reduction in mitochondrial respiration (Figure 3a–d) and membrane potential (Figure 3f), resulting in a decreased OXPHOS ATP production (Figure 3e) in APPswe cells. Mitochondrial oxygen consumption of APPswe cells is altered either under basal conditions and/or when challenged by the mitochondrial modulators, suggesting a less efficient electron transport chain (ETC). Under such conditions, a reduction in mitochondrial membrane potential was observed, hampering the capacity of APPswe cells to produce ATP, which are in accordance with previous observations made in other models carrying APP mutations [79].

Previously, a reduction in mitochondrial membrane potential in AD fibroblasts after thapsigargin-induced Ca^2+^ stress stimulus [80] and the depolarization of the mitochondrial membrane with FCCP inhibit Ca^2+^ uptake were observed [81]. We also found a decrease in glycolysis rate (Figure 3h) and glycolysis capacity in APPswe cells (Figure 3i). Glucose withdrawal prevents glycolysis, rendering cells more dependent on OXPHOS to synthetize ATP. A decrease in glycolysis suggests that APPswe cells are unable to use this pathway to overcome mitochondrial ATP deficits, representing a mal-adaptive response that worsens the cellular energetic status. A similar result was also observed in AD cybrid cell lines that harbor mitochondrial DNA from AD patients platelets [82]. It is described in different cell models that MFN2 depletion changes the cellular metabolic profile, leading to mitochondrial membrane depolarization and decreased cellular oxygen consumption as well as decreased glucose and pyruvate oxidation [83]. Accordingly, a reduction in glucose metabolism was also observed in the neocortex of AD patients that has been correlated with a decreased expression of nuclear and mitochondrial genes that encode proteins of OXPHOS [84]. As previously stated, mitochondria are dynamic organelles that can change morphology and distribution due to fission and fusion events. Mitochondrial fission is regulated by DRP1 and FIS-1 and fusion by MFN1, MFN2, and mitochondrial dynamin like GTPase (OPA1) [85]. Our results obtained in APPswe cells show an impairment in mitochondrial dynamics due to an unbalance between fusion and fission, particularly in MFN2 levels (Figure 4a–e), as well as a slight increase in mitochondrial biogenesis (Figure 4f–h) that did not reach statistical significance, which could represent an attempt to improve mitochondrial metabolic capacity reducing the cascade of deleterious events caused by mitochondrial dysfunction. Accordingly, AD fibroblasts also show defects in mitochondrial dynamics and bioenergetics, reduced ATP generation, and Ca^2+^ dysregulation [80]. Furthermore, abnormal mitochondrial distribution and fragmentation were associated to mitochondrial membrane depolarization and decreased ATP production in M17 cells overexpressing APP [86].

The ER-mitochondria contacts are also involved in the ER stress-mediated cell death and UPR, because several ER co-factors and chaperones are present at MAM. Moreover, impairment in the UPR, apoptosis, and misfolded proteins accumulation are common features of AD [87]. In the early phase of ER stress, the cell adapts to stressful conditions through the increase of ER-mitochondria connection, which facilitates mitochondrial Ca^2+^ uptake and ATP production. However, alterations in MAM functional features, such as Ca^2+^ signaling, induce ER stress and UPR activation, so ER-mitochondria communication could also regulate stress responses and the UPR at different levels [33]. Besides the alterations in ER-mitochondria contacts found in our AD cell model, we also observed that UPR markers are present at MAM and the activation of this stress response occurred through the increase of PERK and IRE1α protein levels (Figure 5c–e). It was previously reported that MFN2 interacts directly with PERK to regulate ER stress-mediated pathways, including UPR, and MFN2 ablation increases PERK activity [88]. PERK is also a member of the MAM that facilitates signals exchanges through these contacts [89] with its upregulation being detected in human post-mortem AD brains [42]. Besides its involvement in UPR, IRE1α stabilizes IP_3_Rs at MAM controlling mitochondrial Ca^2+^ uptake [90]. Under resting conditions, IRE1α deficiency was shown to trigger mitochondrial and energy metabolism alterations [90]. We also found a decrease in the levels of ERO1α (Figure 5f,g), which catalyzes disulfide bond formation in ER during protein folding, and loss of its function leads to misfolded proteins accumulation [44]. This protein also plays a key role in ER redox homeostasis. Under ER stress conditions, the pro-apoptotic transcription factor CHOP can activate ERO1α and promote apoptosis through Ca^2+^ signaling modulation because ERO1α interacts with IP_3_R potentiating Ca^2+^ efflux from ER to mitochondria at MAM [43,91,92]. On the other hand, ERO1α silencing inhibits ER Ca^2+^ release [43,44]. Therefore, the decrease of ERO1α that we observed can lead to the loss of ER redox homeostasis increasing ER stress, simultaneously contributing to prevent Ca^2+^ efflux from ER to mitochondria.

We also observed an increase of PCNA levels in APPswe (Figure 5f,h). This nuclear protein is essential for regulation of DNA replication, repair, and epigenetic alterations; however, PCNA has been associated with cellular events through the formation of a complex with many cytosolic proteins involved in the mitogen-activated protein kinase (MAPK) and phosphatidylinositol 3-kinase (PI_3_K)/protein kinase B (AKT) signaling pathways demonstrating its involvement in cellular signaling [93]. In mature non-proliferating neutrophils, PCNA is located in the cytosol where it serves as a binding platform for procaspases inhibiting their activation and apoptosis, thus regulating neutrophil survival [48]. Furthermore, in APPswe Tg mice, proliferation markers, such as PCNA, seem to induce cell cycle re-entry [94] and its mRNA levels have been found to be increased in brain tissue from AD patients [95] and in APP human neuroblastoma SH-SY5Y cells [1]. Thus, PCNA is an essential co-factor for DNA replication and repair and appears to be involved in pro-survival pathways in response to cellular stress. In the present AD in vitro model, we also observed an accumulation of PCNA in MAM and mitochondria fractions (Figure 5f), supporting its involvement in cellular stress response. 

Internalization of APP and Aβ peptide by mitochondria was observed in vitro [96,97], in human post-mortem AD brains [29,98,99], as well as in APP-overexpressing mice [96,99,100]. It is described that Hsp60 and TOM40, associated to TOM23, are responsible for APP and Aβ internalization into mitochondria [96,97,101]. Moreover, accumulation of APP in the TOM complex affects the mitochondrial import of other proteins and, consequently, impairs mitochondrial functions [102]. As observed in hippocampal neurons of ApoE4 female mice, the increase of the fusion protein MFN1 and TOM40 indicates that the decrease in mitophagy can lead to the accumulation of mitochondrial proteins, due to the accumulation of damaged mitochondria [103]. Immunoblot analyses also revealed that Hsp60 binds to APP and Aβ, and this interaction seems to be increased in the brain of 3xTg-AD mice and in mitochondria isolated from human AD brain [104]. Furthermore, Hsp60 protein levels were increased by ~50% and ~35%, in mitochondrial fractions isolated from the postmortem frontal cortex of SAD and FAD individuals, respectively [104]. In addition, Hsp60 knockdown attenuates APP and Aβ translocation to mitochondria in the neural cell line (C17.2) expressing the APP Swedish mutation [101]. In this study, we also observed an increase in TOM40 (in MAM and mitochondria fractions) (Figure 5f,i) and Hsp60 (in MAM fraction) (Figure 5j,k) in APPswe cells, which can contribute to APP accumulation in mitochondria resulting in mitochondrial dysfunction, as previously described.

In conclusion, our results suggest that the overexpression of the familial APP Swedish mutation causes the accumulation of APP in mitochondria and MAM, which can contribute to the structural and functional changes in these subcellular fractions, affecting the response to stress and inducing energy deficits. Together, our findings support the idea that the defective crosstalk between ER and mitochondria, mediated by MAM, plays a key role in AD physiopathology that may represent a relevant therapeutic target in this disorder.

## Figures and Tables

**Figure 1 biomedicines-09-00881-f001:**
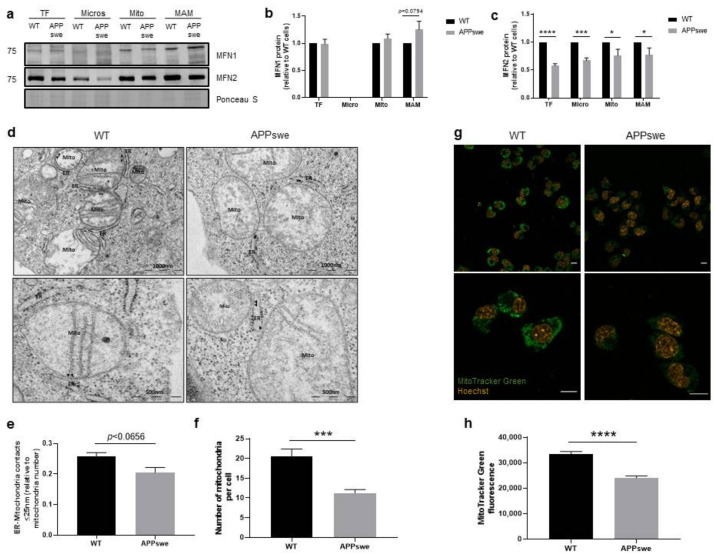
Mitochondria-endoplasmic reticulum (ER) contacts in mouse neuroblastoma cell line (N2A) overexpressing the APP familial Swedish mutation (APPswe) versus wild-type (WT) cells. (**a**–**c**) Representative Western blots and quantification of the ER-mitochondria tethering proteins mitofusin 1 (MFN1) and mitofusin 2 (MFN2) in total fraction (TF), microsomes (Micro), mitochondria (Mito), and mitochondrial-associated membranes (MAMs) (*n* = 5–9). (**d**,**e**) Representative electron micrographs and quantification of ER-mitochondria contacts (MAM) ≤ 25 nm (*n* = 3). (**f**) Profile of mitochondria number per cell (*n* = 3). (**g**,**h**) Representative image of MitoTracker Green (mitochondria, green) and Hoechst (nucleus, yellow) staining and quantification of corrected total fluorescence intensity (CTCF) of stained cells (*n* = 3). Scale bar represents 10 μm. Mito—mitochondria, ER—endoplasmic reticulum, arrow heads—MAM. All data presented as mean ± SEM; *p*-values were obtained by using two-way ANOVA with Sidak’s multiple comparisons test for (**b**,**c**), unpaired *t*-test for (**f**), and non-parametric independent Mann–Whitney U test in (**g**). * *p* ≤ 0.05, *** *p* ≤ 0.001, and **** *p* ≤ 0.0001 were considered significant.

**Figure 2 biomedicines-09-00881-f002:**
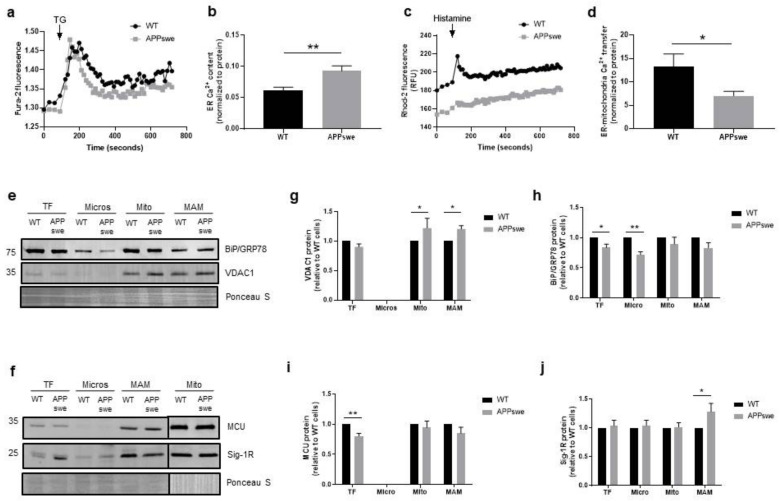
Ca^2+^ transfer from ER to mitochondria in APPswe versus wild-type (WT) N2A cells. (**a**,**c**) Representative traces of cytosolic and mitochondrial Ca^2+^ content evaluated with Fura-2 and Rhod-2 probes, respectively. (**b**,**d**) Quantification of ER Ca^2+^ content after stimulation with 5 µM thapsigargin (TG) in Ca^2+^-free medium using the Fura-2 probe, and ER-mitochondria Ca^2+^ transfer after stimulation with 100 µM histamine in Ca^2+^-free medium using the Rhod-2 probe. Fura-2 and Rhod-2 results represent the average of five or three independent experiments, respectively, performed in triplicate. (**e**–**j**) Representative Western blots and quantification of protein levels of voltage-dependent anion channel 1 (VDAC1) (*n* = 5–9), glucose-regulated protein 78 (BiP/GRP78) (*n* = 5–11), mitochondrial calcium uniporter (MCU) (*n* = 4–8), and sigma-1 receptor (Sig-1R) (*n* = 4–8) proteins in total fraction (TF), microsomes (Micro), mitochondria (Mito), and mitochondrial-associated membranes (MAMs). All data presented as mean ± SEM; *p*-values were obtained by using unpaired *t*-test for (**a**,**c**), two-way ANOVA with Sidak’s multiple comparisons test for (**g**–**i**), and non-parametric Kruskal-Wallis with Dunn’s multiple comparisons test for (**j**). * *p* ≤ 0.05 and ** *p* ≤ 0.01 were considered significant.

**Figure 3 biomedicines-09-00881-f003:**
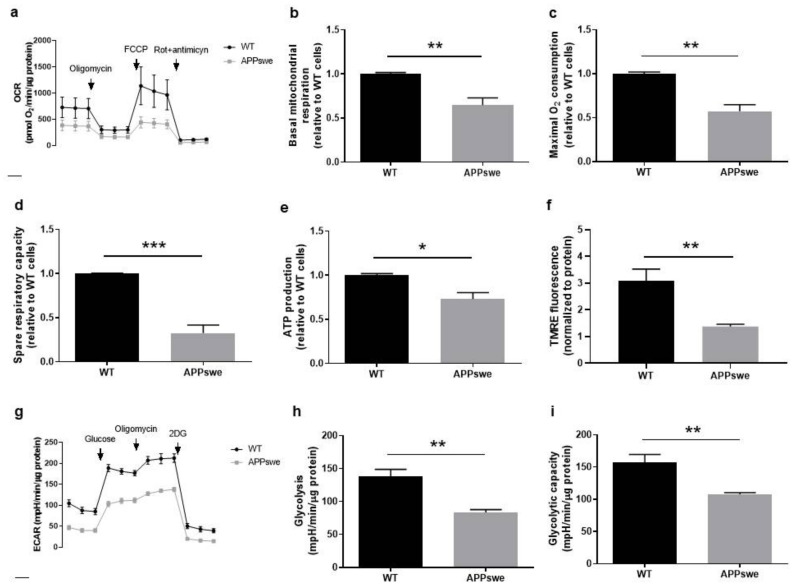
Mitochondrial function and glycolytic activity in APPswe versus wild-type (WT) N2A cells. (**a**) Representative traces of oxygen consumption rate (OCR) evaluated in WT and APPswe cells by sequential addition of 1 µM oligomycin, 1 µM FCCP, and 2 µM rotenone (Rot) plus 2 µM antimycin A by Seahorse analysis (*n* = 4). (**b**–**e**) Quantification of basal mitochondrial respiration, maximal O_2_ consumption, spare respiratory capacity, and ATP production. (**f**) Quantification of mitochondrial membrane potential using the tetramethylrhodamine ethyl ester perchlorate (TMRE) fluorescence probe (*n* = 6). (**g**) Representative traces of extracellular acidification rate (ECAR) evaluated in WT and APPswe cells (average *n* = 4) by sequential addition of 25 mM glucose, 1 µM oligomycin and 100 mM 2-deoxyglucose (2DG) by Seahorse analysis. (**h**,**i**) Quantification of glycolysis rate and glycolytic capacity. Experiments were carried out in quintuplicate for (**a**–**e**,**g**–**i**) and triplicate for (**f**). All data presented as mean ± SEM; *p*-values were obtained by using unpaired *t*-test. * *p* ≤ 0.05, ** *p* ≤ 0.01, and *** *p* ≤ 0.001 were considered significant.

**Figure 4 biomedicines-09-00881-f004:**
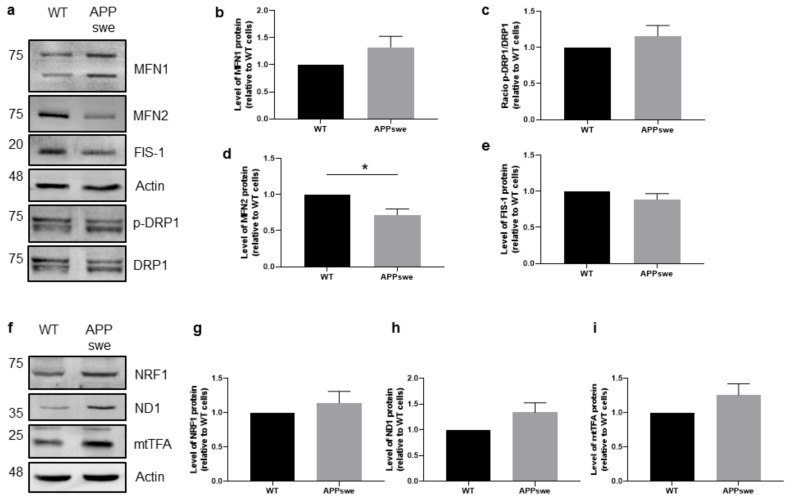
Mitochondrial dynamics and biogenesis in APPswe cells versus wild-type (WT) N2A cells. (**a**–**e**) Representative Western blots and quantification of proteins involved in mitochondrial fusion/fission in total fraction: mitofusin 1 (MFN1) (*n* = 6), phospho-dynamic-1-like protein (*p*-DRP1) (*n* = 6), mitofusin 2 (MFN2) (*n* = 6), and fission-1 (FIS-1) (*n* = 5). (**f**–**i**) Representative Western blots and quantification of proteins involved in mitochondrial biogenesis in total fraction: nuclear respiratory factor 1 (NRF1) (*n* = 6), NADH-ubiquinone oxidoreductase chain 1 (ND1) (*n* = 6), and mitochondrial transcription factor A (mtTFA) (*n* = 6). Values were normalized by actin for all proteins, except for *p*-DRP1 that was normalized by total DRP1. All data presented as mean ± SEM; *p*-values were obtained by using unpaired *t*-test with Welch’s correction. * *p* ≤ 0.05 was considered significant.

**Figure 5 biomedicines-09-00881-f005:**
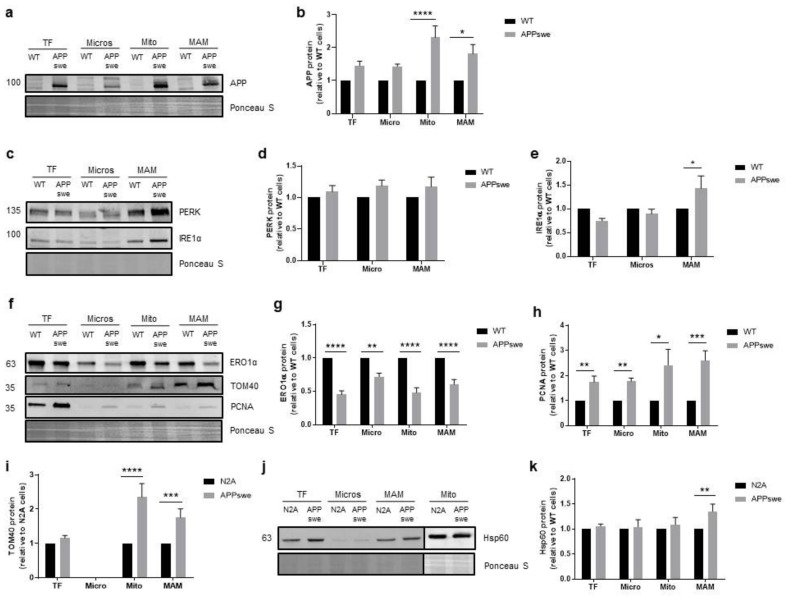
Accumulation of APP and stress responses in APPswe versus wild-type (WT) N2A cells. (**a**,**b**). Representative Western blot and quantification of amyloid β precursor protein (APP) (*n* = 4–6) in total fraction (TF), microsomes (Micro), mitochondria (Mito), and mitochondrial-associated membranes (MAMs). (**c**–**e**). Representative Western blots and quantification of protein kinase R-like ER kinase (PERK) (*n* = 4–5) and inositol-requiring enzyme 1α (IRE1α) (*n* = 3–6) proteins in TF, Micro, and MAM. (**f**–**i**). Representative Western blots and quantification of endoplasmic reticulum oxidase 1 (ERO1α) (*n* = 5–9), proliferating cell nuclear antigen (PCNA) (*n* = 4–10) and outer mitochondrial membrane 40 (TOM40) (*n* = 4–9) proteins in TF, Micro, Mito, and MAM. (**j**,**k**). Representative Western blot and quantification for the chaperone heat shock protein 60 (Hsp60) (*n* = 4–10) in TF, Micro, Mito, and MAM. All data presented as mean ± SEM; *p*-values were obtained by using two-way ANOVA with Sidak’s multiple comparisons test for all proteins, except for (**h**) that was obtained using non-parametric Kruskal-Wallis with Dunn’s multiple comparisons test. * *p* ≤ 0.05, ** *p* ≤ 0.01, *** *p* ≤ 0.001, and **** *p* ≤ 0.0001 were considered significant.

**Table 1 biomedicines-09-00881-t001:** Primary antibodies utilized for Western blot.

Primary Antibody	Dilution	Species	Company	Catalog Number	Location
APP	1:4000	Rabbit	Sigma-Aldrich	A8717	St. Louis, MO, USA
β-Actin	1:10,000	Mouse	Sigma-Aldrich	A5316	St. Louis, MO, USA
BiP/GRP78	1:1000	Mouse	BD Transduction	610978	San Jose, CA, USA
DRP1	1:1000	Rabbit	Cell Signaling	8570	Danvers, MA, USA
ERO1α	1:1000	Mouse	Santa Cruz Biotechnology	sc-100805	Santa Cruz, CA, USA
FIS-1	1:500	Rabbit	Novus Biologicals	NB100-56646	Littleton, CO, USA
Hsp60	1:1000	Mouse	BD Transduction	611563	San Jose, CA, USA
IRE1α	1:500	Rabbit	Cell Signaling	3294	Danvers, MA, USA
MCU	1:1000	Rabbit	Cell Signaling	14997	Danvers, MA, USA
Mfn1	1:1000	Rabbit	Santa Cruz Biotechnology	sc-50330	Santa Cruz, CA, USA
Mfn2	1:1000	Mouse	Santa Cruz Biotechnology	sc-100560	Santa Cruz, CA, USA
mtTFA	1:1000	Goat	Santa Cruz Biotechnology	sc-23588	Santa Cruz, CA, USA
ND1	1:1000	Goat	Santa Cruz Biotechnology	sc-20493	Santa Cruz, CA, USA
NRF-1	1:1000	Rabbit	Santa Cruz Biotechnology	sc-33771	Santa Cruz, CA, USA
PCNA	1:1000	Mouse	Santa Cruz Biotechnology	sc-25280	Santa Cruz, CA, USA
PERK	1:500	Rabbit	Cell Signaling	3192	Danvers, MA, USA
p-DRP1	1:500	Rabbit	Cell Signaling	3455	Danvers, MA, USA
Sigma1R	1:2000	Goat	Santa Cruz Biotechnology	sc-22948	Santa Cruz, CA, USA
Tom40	1:1000	Mouse	Santa Cruz Biotechnology	sc-365467	Santa Cruz, CA, USA
VDAC1	1:1000	Mouse	Santa Cruz Biotechnology	sc-390996	Santa Cruz, CA, USA

**Table 2 biomedicines-09-00881-t002:** Secondary antibodies utilized for Western blot.

Secondary Antibody	Dilution	Species	Company	Catalog Number	Location
IgG (anti-goat)	1:10,000	Rabbit	Santa Cruz Biotechnology	sc-2771	Santa Cruz, CA, USA
IgG (anti-mouse)	1:10,000	Goat	Thermo Fisher	31320	Waltham, MA, USA
IgG (anti-rabbit)	1:20,000	Goat	GE Healthcare	NIF1317	Chicago, IL, USA

## Data Availability

All data are available upon contact of corresponding authors.

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
