# Peer review of "Structural and Functional Alterations in Mitochondria-Associated Membranes (MAMs) and in Mitochondria Activate Stress Response Mechanisms in an In Vitro Model of Alzheimer’s Disease"

_biomedicines, 2021, doi:10.3390/biomedicines9080881_

Round 1
Reviewer 1 Report
The manuscript is well-written and the results are well-discussed. I have some suggestions to improve the quality of the manuscript.
Please, define all acronymous the first time they are cited in the manuscript and the acronymous must be specified only the first time it is cited.
I suggest to not anticipate the results of the study in the introduction.
The material and methods section should start by drscfibin the study design.
Since in this study Authors show several results, I suggest to summarize the main findings in a a bulleted list and then to discuss them.
I suggest to separate the paragraph of the conclusion.
Author Response
The manuscript is well-written and the results are well-discussed. I have some suggestions to improve the quality of the manuscript.
Point 1: Please, define all acronymous the first time they are cited in the manuscript and the acronymous must be specified only the first time it is cited.
Response 1: We thank the comment and we have corrected this situation in the revised version of the manuscript.
Point 2: I suggest to not anticipate the results of the study in the introduction.
Response 2: We thank the comment and, in the introduction section of the revised version of the manuscript, we have removed the brief description of the results.
Point 3: The material and methods section should start by describe the study design.
Response 3: We agree with the comment and we have added a description of the study design in the material and methods section (lines 85-91) in the revised manuscript.
Point 4: Since in this study Authors show several results, I suggest to summarize the main findings in a bulleted list and then to discuss them.
Response 4: We thank and agree with the comment. We have added a summary of the main results in discussion section (lines 532-537) in the revised manuscript.
Point 5: I suggest to separate the paragraph of the conclusion.
Response 5: We thank the comment and the conclusion paragraph have been separated in discussion section in the revised version of the manuscript.

Reviewer 2 Report
Pereira et al. have conducted an in-depth study of the role of MAM (mitochondria associated membranes) in APPswe Alzheimer’s disease model and performed a wide array of tests examining all the relevant parameters and providing comprehensive data on the matter. The function and molecular biology of MAM is an actively investigated topic and it is of utmost importance for furthering our understanding of the pathogenesis of the Alzheimer’s disease. This work is highly informative and thorough and we recommend this article for publication in your journal. However, we would like to provide some comments regarding particular sections of the text:
- The authors have measured average mitochondrial transmembrane potential and rate of respiration and found considerable decrease of these parameters in the APPswe cells which suggests suppressed mitochondrial activity. As far as we understand, the data in these experiments were normalized to total cellular protein. However, authors also report a decrease in total mitochondrial content as measured by Mitotracker Green fluorescence. Thus, the decrease in respiration rate and transmembrane potential might be, at least in part, caused not by biochemical alterations but by decreased mitochondrial biomass. We suggest that this caveat is discussed.
- During the respirometry, several parameters were measured, such as basal respiration, maximal respiratory capacity, and spare respiratory capacity, but the meaning of each of these parameters in the context of the experiment is not provided and respirometry results are reported rather briefly. In our opinion, it would add to the clarity of the article if the relevant section is expanded with some elaboration about these parameters.
- Authors report decreased Ca flux from ER to mitochondria (lines 324-325, 345-348). Several other sources indicate that AD is characterized by mitochondrial Ca overload [Núñez et al. 2008; Esteras & Abramov, 2020; Calvo-Rodriguez & Bacskai, 2020]. Adding discussion of this contradiction would add to the comprehensiveness of the article.
- Please clarify the rationale behind the 25nm distance being the cutoff for MAM detection in TEM microphotographs (lines 161-162) or provide a reference.
Author Response
Pereira et al. have conducted an in-depth study of the role of MAM (mitochondria associated membranes) in APPswe Alzheimer’s disease model and performed a wide array of tests examining all the relevant parameters and providing comprehensive data on the matter. The function and molecular biology of MAM is an actively investigated topic and it is of utmost importance for furthering our understanding of the pathogenesis of the Alzheimer’s disease. This work is highly informative and thorough and we recommend this article for publication in your journal. However, we would like to provide some comments regarding particular sections of the text:
Point 1: The authors have measured average mitochondrial transmembrane potential and rate of respiration and found considerable decrease of these parameters in the APPswe cells which suggests suppressed mitochondrial activity. As far as we understand, the data in these experiments were normalized to total cellular protein. However, authors also report a decrease in total mitochondrial content as measured by Mitotracker Green fluorescence. Thus, the decrease in respiration rate and transmembrane potential might be, at least in part, caused not by biochemical alterations but by decreased mitochondrial biomass. We suggest that this caveat is discussed.
Response 1: We thank the comment and we have made reference in the revised manuscript to the fact that the decreased respiration rate and transmembrane potential observed might be caused, at least in part, by the reduced mitochondrial biomass (results, section 3.3, lines 410-412).
Point 2: During the respirometry, several parameters were measured, such as basal respiration, maximal respiratory capacity, and spare respiratory capacity, but the meaning of each of these parameters in the context of the experiment is not provided and respirometry results are reported rather briefly. In our opinion, it would add to the clarity of the article if the relevant section is expanded with some elaboration about these parameters.
Response 2: We thank the comment and we have merged points 3.3 (referent to respiratory parameters) and 3.4 (glycolysis parameters) in the results section. We added information to clarify the meaning of the respiratory (lines 375-387) and glycolysis (lines 413-425) parameters, both in the results and in the discussion sections (lines 625-630 and 634-639) of the revised manuscript.
Point 3: Authors report decreased Ca flux from ER to mitochondria (lines 324-325, 345-348). Several other sources indicate that AD is characterized by mitochondrial Ca overload [Núñez et al. 2008; Esteras & Abramov, 2020; Calvo-Rodriguez & Bacskai, 2020]. Adding discussion of this contradiction would add to the comprehensiveness of the article.
Response 3: We thank the suggestion. In the discussion section, we have mentioned the increase of mitochondrial Ca2+ described in an AD mice model by Calvo-Rodrigues and colleagues. Moreover, in the revised manuscript we have mentioned additional studies supporting mitochondrial Ca2+ overload in AD and a discussion of the contradictory findings concerning decreased Ca2+ flux from ER to mitochondria that we and others show and the mitochondrial Ca2+ overload observed by others (lines 561-567).
Point 4: Please clarify the rationale behind the 25nm distance being the cutoff for MAM detection in TEM microphotographs (lines 161-162) or provide a reference.
Response 4: Thanks for the suggestion. We have added information to clarify the cutoff at 25 nm for MAM detection in TEM and provide a reference (lines 165-166) in the material and methods section (point 2.4) in the revised version of the manuscript.